# Efficacy of Home-Based Transcranial Direct Current Stimulation on Experimental Pain Sensitivity in Older Adults with Knee Osteoarthritis: A Randomized, Sham-Controlled Clinical Trial

**DOI:** 10.3390/jcm11175209

**Published:** 2022-09-02

**Authors:** Geraldine Martorella, Kenneth Mathis, Hongyu Miao, Duo Wang, Lindsey Park, Hyochol Ahn

**Affiliations:** 1College of Nursing, Florida State University, Tallahassee, FL 32306, USA; 2McGovern Medical School, The University of Texas Health Science Center, Houston, TX 77030, USA; 3Department of Statistics, Florida State University, Tallahassee, FL 32306, USA

**Keywords:** knee osteoarthritis, pain, quantitative sensory testing, conditioned pain modulation, transcranial direct current stimulation

## Abstract

Although transcranial direct current stimulation (tDCS) is encouraging regarding clinical pain intensity for individuals with knee osteoarthritis, very few studies have explored its impact on experimental pain sensitivity, which may hinder our understanding of underlying therapeutic mechanisms. The purpose of this study was to assess the efficacy of 15 home-based tDCS sessions on experimental pain sensitivity and explore its relationships with clinical pain intensity. We randomly assigned 120 participants to active tDCS (*n* = 60) and sham tDCS (*n* = 60). Quantitative sensory testing (QST) was used, including heat pain threshold and tolerance, pressure pain threshold, and conditioned pain modulation. Patients in the active tDCS group exhibited reduced experimental pain sensitivity as reflected by all QST measures at the end of treatment. Furthermore, correlations were observed between changes in clinical pain intensity and experimental pain sensitivity. These findings warrant further studies on tDCS and experimental pain sensitivity in patients with knee osteoarthritis and exploring the magnitude and sustainability of effects on a longer term.

## 1. Introduction

Knee osteoarthritis (OA) is the most common form of arthritis [1]. Severe joint pain, the chief complaint when people seek treatment, compromises quality of life [2]. The etiology of pain related to knee OA is still a phenomenon under investigation. While peripheral sensitization became rapidly evident based on the inflammatory process of the joint, central sensitization emerged more recently and could explain the ongoing debate about the variable association between clinical pain intensity and radiographic knee OA evidence or the presence of pain after total knee replacement in some individuals [3,4]. Numerous studies have examined altered pain processing and central sensitization patterns in individuals with knee OA, suggesting that central hyperexcitability and pain sensitization is present and that it is related to symptom severity [5].

The recent emphasis on central sensitization in this population has brought to light interventions targeting the central nervous system, such as transcranial direct current stimulation (tDCS), both in clinic-based and home-based settings and led to investigations of experimental pain sensitivity. Indeed, experimental pain sensitivity defined as “the result of complex processes at different sites of the pathways that are involved in the generation, transmission, elaboration and perception of pain” [6] in response to a standardized and controlled stimulus can help illuminate nociceptive processes and understand individual variability as to why a stimulus is perceived as more or less painful. Our work with individuals suffering from knee OA has previously shown that both clinic-based and home-based tDCS can lead to a reduction in clinical pain intensity [7,8]. However, although individuals with OA pain have demonstrated higher levels of experimental pain sensitivity [5,9], few studies have evaluated the impact of tDCS on experimental pain sensitivity, which might prevent from clarifying underlying therapeutic mechanisms. Additionally, when targeting chronic pain, tDCS has been mostly used with the anode and cathode electrodes, respectively, placed over the primary motor cortex (M1) and the supraorbital region (SO) [10,11,12], but this configuration needs further validation in relation to specific chronic pain conditions such as OA.

We were the first to validate, through a sham-controlled pilot study, the impact of clinic-based tDCS on experimental pain sensitivity and thus on central pain processing in the context of knee OA [13]. Furthermore, we were able to observe the association between positive changes in experimental pain sensitivity and reduction in clinical pain intensity in this population. We then decided to determine if similar benefits could be obtained using home-based tDCS without the controlled environment of a clinic. Our open-label pilot study showed the preliminary benefits of home-based tDCS on experimental pain sensitivity [14] along with our randomized controlled pilot trial combining home-based tDCS with a mindfulness-based approach [15]. Building on these initial findings, examining the efficacy of home-based tDCS on clinical pain intensity, experimental pain sensitivity, and their relationship in a large-scale controlled trial may strengthen our understanding in uncovering the underlying mechanism of action. In the primary study focused on clinical pain [7], we observed a significant reduction of pain intensity (Cohen′s d 1.20; *p*-value < 0.0001), while participants expressed a high level of satisfaction and no adverse events.

Thus, the purpose of this study was (1) to determine the efficacy of a three-week, home-based tDCS treatment on experimental pain sensitivity in older adults with symptomatic knee OA and (2) to examine the relationship between experimental pain sensitivity and OA-related clinical pain intensity. Our hypothesis was that home-based tDCS would decrease experimental pain sensitivity more than sham tDCS and that a decrease in experimental pain sensitivity would be associated with a decrease in clinical pain intensity.

## 2. Materials and Methods

### 2.1. Design

The protocol has been registered at www.clinicaltrials.gov (accessed on 10 July 2022) (NCT04016272). After the ethical approval was granted from the Institutional Review Board (IRB), we proceeded to a double-blind, randomized, sham-controlled, phase II parallel-group study with two groups (sham and active tDCS).

Data collection was completed at baseline (T0) and at the end of each week of treatment (T1, T2, T3). All participants competed a sociodemographic questionnaire. Medical history was also assessed, including age, sex, height, weight, duration of OA, current and past treatments for knee OA < comorbid conditions, and current medications.

### 2.2. Participants

Our detailed protocol and enrollment procedures are described in a previous article reporting the primary outcomes of our study [7]. Summarily, after signing a consent form, 120 individuals were randomly assigned to two groups (i.e., 60 participants in each group). Similar to our previous work [8], participants were considered eligible if they (1) had symptomatic knee OA based on American College of Rheumatology Clinical criteria [16], (2) had had knee OA pain in the past 3 months with an average of at least 30 on a 0–100 numerical rating scale (NRS) for pain, (3) could speak and read English, and (4) had no plan to change pain-related medication during the entire trial. According to American College of Rheumatology criteria [16], participants should meet at least 3 of 6 criteria, including age > 50 years, stiffness < 30 min, crepitus, bony tenderness, bony enlargement, and no palpable warmth. The age range of 50–85 years was selected to include a higher proportion of persons with knee OA pain whose pain sensitivity has not changed with advancing age [17]. Participants were excluded if they had any concurrent medical conditions that could bias the interpretation of findings (e.g., fibromyalgia or rheumatoid arthritis), posed a safety risk for any of the assessment or intervention procedures (e.g., seizure, stroke, or brain tumor), or precluded the successful execution of the study (e.g., cognitive problems or substance abuse).

### 2.3. Interventions and Procedures

The two types of interventions were active tDCS and sham tDCS. Active tDCS delivered a 2 mA electrical current for 20 min, and sham tDCS consisted of delivering an active stimulation (i.e., 2 mA electrical current) for a few seconds to mimic the sensations of active tDCS and maintain blinding of participants.

Participants were given brief instructions on the procedure and use of the device. Evidence-based guidelines recommend 20 min M1-SO stimulation using 2 mA electrical current intensity for potential efficacy among individuals suffering from chronic pain [12]. tDCS with a constant current intensity of 2 mA was applied for 20 min five times a week (Monday to Friday) for 3 weeks via the Soterix 1 × 1 tDCS mini-CT Stimulator device (Soterix Medical Inc., New York, NY, USA; 6.5 inches long, 3 inches wide, 0.7 inches thick) with headgear and 5 × 7 cm sponge electrodes. This single-position headgear with labeled sponge markers eliminated room for incorrect placement. Participants could only administer a stimulation session after being provided with a single-use unlock code by the research staff once proper contact quality was achieved (only the on/off button was activable by the study participants). After entering the code, participants could see the timer showed on the screen, and the device turned off automatically at the end of the session. For sham stimulation, the electrodes were placed similarly as for active stimulation, while a 2 mA current was delivered for only 30 s. This sham stimulation method has been shown to be reliable and imperceptible from active treatment [18,19].

### 2.4. Outcomes

#### 2.4.1. Clinical Pain Intensity

Clinical pain measures are described in detail in our previous article [7]. Briefly, clinical pain intensity (i.e., knee pain) over the past 24 h was assessed using a Numeric Rating Scale (NRS), with anchors being 0 (no pain) to 100 (worst pain imaginable) [20,21].

#### 2.4.2. Experimental Pain Sensitivity

A multimodal Quantitative Sensory Testing (QST) battery was administered for experimental pain sensitivity: heat pain threshold (HPTh), heat pain tolerance (HPTo), pressure pain threshold (PPT), and conditioned pain modulation (CPM). These measures were taken using equipment, including a Medoc TSA-II Neurosensory Analyzer (Medoc Ltd., Ramat Yishai, Israel) and Wagner pressure algometer (Wagner, Greenwich, CT, USA). QST has demonstrated moderate reliability in relation to neural sensitivity [22] and has been underlined for its promise regarding phenotyping knee OA pain and predicting outcomes after total knee replacement [23]. Particularly, CPM is a psychophysical experimental measure reflecting the endogenous pain inhibitory pathway (i.e., descending pain inhibition) also known as the ”pain inhibits pain” paradox [24], which is reliable [25] and has shown associations with pain patterns and duration of symptoms in individuals with knee OA [26,27].

Thermal stimuli were delivered to measure heat pain threshold and heat pain tolerance at the knee using an ascending method of limits. From the baseline of 32 °C, the thermode temperature increased at a rate of 0.5 °C per second until the participants pressed a button to stop heat stimuli. Participants were asked to press the button when the sensation “first becomes painful” to assess the HPTh and when they “no longer feel able to tolerate the pain” to assess the HPTo. The average of the three trials was computed to determine HPTo and HPTh. Then, PPT at the knee was measured by using blunt mechanical pressure delivered via a digital pressure algometer. Increasing pressure was applied continually (rate of 0.3 kgf/cm^2^/s), while participants were instructed to notify the experimenter when the sensation “first becomes painful” to assess the PPT. The results of the three trials were averaged to determine PPT.

Ten minutes after assessing PPT, CPM, which reflects pain inhibition, was assessed [28] by calculating the change in PPT (trapezius) immediately after immersing the contralateral hand in cold water (12 °C) for one minute [13]. The water was maintained at a constant temperature and constantly circulated to prevent warming around the immersed hand. An increase in PPT after cold water immersion indicates pain inhibition.

#### 2.4.3. Sample Size

This study recruited and randomly assigned 120 individuals to two groups (60 participants per group). Considering a 10% attrition rate, the sample size was found sufficient to achieve an 80% power to detect an effect size of 0.54 or higher at a significance level of 0.05.

#### 2.4.4. Statistical Analysis

All statistical analyses were performed using R Statistical Software (version 4.1.2; R Foundation for Statistical Computing, Vienna, Austria). Statistical significance was achieved based on α = 0.05, and Bonferroni correction was not adopted as practiced in previous QST studies. Raw data were examined for missing values and outliers first, and Shapiro’s test was used to check the normality assumption. *t*-Tests were then performed to compare the 3-week score changes between the active and the sham groups for normally distributed HPTh and PPT, respectively; and Wilcoxon rank-sum tests were used to compare the 3-week score changes of non-normally-distributed HPTo and CPM between the active and the sham groups. In addition, based on linear mixed-effects model, profile analysis was conducted to compare trends over time of the QST score changes between groups. Generalized estimating equation (GEE) model allowed to examine the relationships between the longitudinal observations of the four QST score changes (from baseline to 1 week, 2 weeks, 3 weeks) and the same group of potential confounding variables previously mentioned. Moreover, to investigate the outcome differences in relation to sex, we performed the sex-stratified analysis. Finally, Spearman rank-order correlation was used to test the association between the 3-week changes in each of the four QST measures and the 3-week changes in clinical pain intensity (NRS).

## 3. Results

### 3.1. Participants

In total, 120 participants were successfully recruited and randomly assigned to the active tDCS group (60 subjects) or the sham tDCS group (60 subjects), and the sample included approximately 68% women. CONSORT flow diagram as provided in Martorella et al. (2022) [7]. See Table 1 in Martorella et al. (2022) [7] for the baseline demographics and clinical characteristics of all participants in this study. Briefly, the age distributions of the sham tDCS group (66.6 ± 8.4 years) and the active tDCS group (65.3 ± 8.4 years) were not statistically significantly different. Similarly, the gender, race, and marital status distributions of the two groups were not significantly different. The BMI of the sham tDCS group was 32.5 ± 8.3 kg/m^2^, and it was 32.7 ± 8.7 kg/m^2^ for the active tDCS group. The OA duration was 69.3 ± 82.9 months and 71.4 ± 75.9 months for the sham and the active tDCS groups, respectively. Detailed baseline QST characteristics were presented in Table 1.

### 3.2. Experimental Pain Sensitivity

Among the four QST measures, HPTh and PPT were found significant following the Gaussian distribution, while HPTo and CPM were not. For the 3-week changes, HPTh (Cohen′s d = 0.69; *p*-value < 0.01), PPT (Cohen′s d = 1.39; *p*-value < 0.0001), HPTo (Cohen′s d = 0.42; *p*-value = 0.03), and CPM (Cohen′s d = 0.53; *p*-value < 0.01) were all found significantly different between the active and the sham tDCS groups (see Table 2). The average increase in HPTh at 3 weeks from baseline was 1.13 ± 3.43 for the active group, while it decreased by1.13 ± 3.08 for the sham group. The average increase in PPT at 3 weeks from baseline was 0.56 ± 0.52 for the active group, while there was a decrease of 0.30 ± 0.70 for the sham group. The HPTo increased by 0.84 ± 2.10 at 3 weeks from baseline for the active group and decreased by 0.10 ± 2.39 for the sham group. At 3 weeks from baseline, the CPM increased by 0.14 ± 0.67 for the active group and decreased by 0.21 ± 0.65 for the sham group (see Figure 1).

Profile analysis results suggested that significant interactions between the groups and time factors existed for all the four QST measures; that is, the treatment group has a statistically significantly different trend over time from that of the sham group for each of the four QST measures. After regressing each of the four QST measure changes over time (week 1, week 2, week 3 from baseline) against selected covariates using GEE models, we found that (1) for the changes of HPTh, treatment group (*p*-value < 0.01), age (*p*-value = 0.03), sex (*p*-value < 0.0001), race for Hispanic or Latino (*p*-value = 0.02), and baseline HPTh (*p*-value < 0.0001) were statistically significant; (2) for the changes of PPT, treatment group (*p*-value < 0.0001), age (*p*-value < 0.0001), race for American Indian or Alaska Native (*p*-value = 0.01) and Black African American (*p*-value = 0.01), and baseline PPT (*p*-value < 0.0001) were found significant. However, the result for American Indian or Alaska Native was not reliable due to the small sample size. (3) For the changes of CPM, the treatment group (*p*-value < 0.0001), the baseline PPT (*p*-value = 0.04), and baseline CPM (*p*-value < 0.0001) were found significant. (4) No significant covariates were found with respect to the change of HPTo.

Based on the sex-stratified analysis, the changes of HPTh at 3 weeks from baseline were found significantly different between females and males (*p*-value < 0.0001), and the multivariate regression results suggested that treatment group, age, and baseline NRS were significant for males only with respect to the 3-week changes of HPTh. However, we did not find statistically significant differences for the changes of PPT and CPM between males and females in both groups. When analyzing the data of males and females separately, we found that (1) for the changes of PPT, baseline NRS and baseline PPT were found significant for the male group, and baseline PPT was found significant for the female group; (2) for CPM changes, age, choice of the knee, baseline NRS, and baseline HPTh were found significant for the male group, and baseline CPM was significant for the female group.

### 3.3. Relationship between Clinical Pain Intensity and Experimental Pain Sensitivity

We found the significant associations between the changes in the experimental pain sensitivity measures and the changes in clinical pain intensity (NRS) using the Spearman rank-order correlation: HPTh (correlation coefficient ρ = −0.20, *p*-value = 0.03), HPTo (correlation coefficient ρ = −0.26, *p*-value < 0.01), and PPT (correlation coefficient ρ = −0.34, *p*-value < 0.001). However, although the correlation between CPM and NRS changes showed a trend towards significance, it was not found to be significant (correlation coefficient ρ = −0.16, *p*-value = 0.08).

## 4. Discussion

This is the first randomized, sham-controlled trial to investigate the efficacy of home-based tDCS on experimental pain sensitivity in older adults suffering from knee OA. We revealed that 15 sessions of home-based tDCS reduced experimental pain sensitivity with moderate to large effect sizes. Additionally, we observed an association between improvements in experimental pain sensitivity and clinical pain-intensity reduction.

Our results converge with prior studies on tDCS for individuals with knee OA that showed an increase in pain tolerance and threshold, as reflected by HPTh, HPTo, PPT, and CPM, with clinic-based tDCS [13,29]. In this study, we show that self-administered tDCS impacts experimental pain sensitivity for patients with knee OA, which highlights the fidelity of delivering tDCS in the home setting and supports its future implementation. Additionally, by looking at the trend over time, we observed an increase in CPM. This result is aligned with the results from Tavares and colleagues, who recorded significant effects with 15 sessions of the same tDCS approach in patients with knee OA [29]. Additionally, CPM at baseline, reflecting endogenous pain-modulation capacity, accounted for CPM changes in females but not males. These results are in concordance with previous results suggesting sex differences in central sensitization and descending pain inhibition in patients with knee OA and the need to consider these in the development and evaluation of therapeutic approaches [27,30].

Moreover, as observed previously with clinic-based tDCS [13], we showed that beneficial changes in experimental pain sensitivity are associated with less clinical pain intensity for home-based tDCS as well. Nevertheless, a meta-analysis reported that, although tDCS improved pain intensity, it did not improve pain threshold and tolerance, i.e., experimental pain sensitivity, in healthy subjects [31]. This can be explained by variations in the current intensity and polarity used across studies. In addition, it suggests that the relationship observed between the improvement of experimental pain sensitivity and the reduction in clinical pain intensity in knee OA patients is likely to be related to the mitigation of central sensitization and neuroplasticity and of its impact on endogenous central pain inhibition [32], which is absent in healthy subjects. These findings further uncover the underlying mechanism of tDCS and could explain the association between improvements in experimental pain sensitivity and clinical pain intensity seen in patients with knee OA. Notably, a study that looked specifically at the impact of tDCS on patients with knee OA and impaired descending pain inhibition found similar benefits [29].

Some limitations in this study can be underlined. First, we did not collect the follow-up measures for experimental pain sensitivity, so we did not explore sustained effects of tDCS on experimental pain sensitivity. This hinders our capacity to assess its clinical value for chronic conditions that are often refractory to treatment. Moreover, although we focused on the relationship between clinical pain intensity and experimental pain sensitivity, we did not measure any physiological biomarkers, such as brain-derived neurotrophic factor (BDNF), β-endorphin, and TNF-α, which could advance our understanding of tDCS’s underlying mechanism in the context of knee OA. While still embryonic, research suggests that BDNF levels, for instance, could highlight pain inhibition and neuroplastic changes, thus illustrating treatment response [33]. Of note, we previously found a relationship between BDNF, heat pain threshold, and clinical pain (numeric rating scale) in individuals with knee OA [34,35].

Our study provides insights for future research. Adding some maintenance tDCS sessions seems to be a favorable approach, as research suggests that repeated sessions may induce neuroplastic changes, thus potentially leading to long-lasting benefits [36]. Of note, a previous study did not find any sustained effects on both clinical and experimental pain measures in individuals with knee OA two months after 15 tDCS sessions [29]. However, the protocol focused on subjects with pre-identified dysfunctional descending pain inhibitory system (based on CPM assessment), suggesting that this subgroup may benefit from extended tDCS therapy but also underlining the importance of subject selection or responder discrimination in examining tDCS effects [32]. Heterogeneity of samples can be very high in the context of pain with, for instance, individuals that are opioid-naïve as opposed to chronic opioid users or a variability in CPM. Future research on physiological biomarkers and response predictors will help uncovering underlying mechanisms and fine-tuning tDCS treatment approach [36].

## 5. Conclusions

Fifteen home-based tDCS (M1-SO) sessions were effective in improving experimental pain sensitivity and endogenous pain inhibition, leading to clinical pain improvement. These findings warrant further multi-site, large-scale research exploring these benefits on subgroups of patients suffering from knee OA (e.g., sex) and their maintenance.

## Figures and Tables

**Figure 1 jcm-11-05209-f001:**
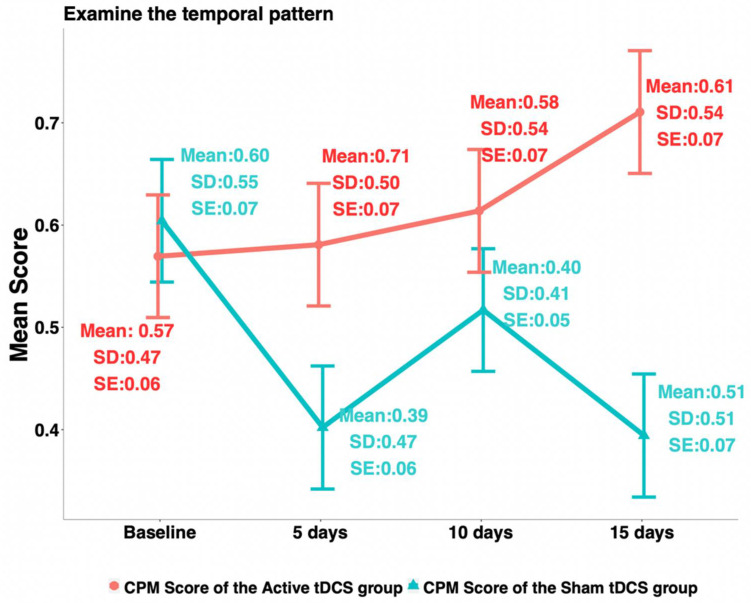
Trends of conditioned pain modulation.

**Table 1 jcm-11-05209-t001:** Baseline QST characteristics of the participants.

Variable	Sham tDCS (*n* = 60)	Active tDCS (*n* = 60)	*p*-Value
HPTh, M(SD)	40.17 (3.42)	39.40 (3.30)	0.12
HPTo, M(SD)	45.06 (3.17)	44.85 (2.74)	0.51
PPT, M(SD)	2.57 (1.17)	2.40 (1.02)	0.57
CPM, M(SD)	0.60 (0.55)	0.57 (0.47)	0.58

Note. M, mean; SD, standard deviation; HPTh, heat pain threshold; HPTo, heat pain tolerance, PPT, pressure pain threshold; CPM, conditioned pain modulation.

**Table 2 jcm-11-05209-t002:** Comparison between groups on changes from baseline QST measures.

Variable	Sham Group (*n* = 60)	Active Group (*n* = 60)	Effect Size (d)	*p*-Value
HPTh Change	−1.13 ± 3.08	1.13 ± 3.43	0.69	<0.01
HPTo Change	−0.10 ± 2.39	0.84 ± 2.10	0.42	0.03
PPT Change	−0.30 ± 0.70	0.56 ± 0.52	1.39	<0.0001
CPM Change	−0.21 ± 0.65	0.14 ± 0.67	0.53	<0.01

Note. Mean ± standard deviation is presented in the first two columns. HPTh, heat pain threshold; HPTo, heat pain tolerance; PPT, pressure pain threshold; CPM, conditioned pain modulation.

## Data Availability

Not applicable.

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
