# Peer review of "Efficacy of Home-Based Transcranial Direct Current Stimulation on Experimental Pain Sensitivity in Older Adults with Knee Osteoarthritis: A Randomized, Sham-Controlled Clinical Trial"

_jcm, 2022, doi:10.3390/jcm11175209_

Round 1

Reviewer 1 Report

The authors present a secondary analysis from an RCT examining home based tDCS for OA related knee pain focused on pain sensitivity.  The methods are well described and supported with the introduction.  Analytic plan is well described and presented. It may be helpful to state the primary paper outcomes for reference as they are relevant to interpretation of the current paper. The authors note they exist but do not provide a summary. The discussion is well presented and balanced.

One issue to address is the last result presented is referenced as "marginally significant" rather than a trend.  Since it did not reach significance it should be described either as non-significant OR a trend.

Reviewer 2 Report

Introduction

1. The introduction is well written, clearly showing why this study was undertaken. However, it might be good to say a little about what experimental pain sensitivity is.

2. Was there any specific interest for focusing this RCT on older adults, instead of simply knee OA patients? If any, it would be good to explain this in the introduction.

Methods

3. This study is reported as a “double-blind, randomized, sham-controlled” trial (i.e., a phase III clinical trial). However, in the manuscript, as well as on CT.gov, it has been identified as a “phase II” trial. Please note that phase II clinical trials are rather dose-finding trials.

4. On CT.gov, the study is still under the “Recruiting” status. How do the authors explain this?

5. The authors reported that “participants who were 50-85 years old were considered eligible”. Therefore, why was this study titled as being a study on older adults, as it potentially included people under 60 years old?

6. Please change the title “tDCS and sham conditions” to “Interventions and procedures”, then add a first paragraph briefly describing active tDCS and sham tDCS just before the current text under “tDCS and sham conditions”.

7. The reviewer suggests moving the paragraph on “Data collection” to the “Design” subsection, as a second paragraph to that subsection, then deleting the title “Data collection”.

8. To ease understanding to future readers, the reviewer suggests adding a subtitle “Outcomes”, putting “Clinical pain intensity” and “Experimental pain sensitivity” under this new subtitle.

9. In accordance with the CONSORT 2010 checklist (http://www.consort-statement.org/), please add a subtitle on “Sample size”, explaining how sample size was determined, as well as a subtitle on “Randomisation and blinding”. The reviewer understands that the detailed protocol has been described elsewhere; however, it is important to say a little about these in the current manuscript.

10. In the Statistical analysis section, the authors stated that “Bonferroni correction was not adopted as practiced in previous QST studies”. The why of this needs to be explained. In sentence 2, is “Unless otherwise specified” necessary (i.e., was any other threshold considered)? On line 165, please change “to quantify” to “to test”.

Results

11. Please consider providing a CONSORT Flow Diagram for this research.

12. Table 1 reporting the baseline characteristics of participants should include more variables. Please consider adding summary statistics for variables including age, sex, BMI, duration of knee OA, OA grade (K-L), current and past treatments for knee OA, comorbid conditions.

13. Table 2 reports comparison between groups on changes from baseline QST measures, but the timepoint considered relative to baseline is not reported in the title. The reviewer wonders if a graph showing these comparisons from baseline to all timepoints (with p-values for between-group differences) would not be more informative. Such graph may then be provided for each main outcome.

14. There is no mention of safety results. Please consider adding a subtitle on this important issue. Otherwise, explain why safety data were not collected.

Discussion

15. There is a double report of the conclusion of the study (at the end of the discussion, and in title 5). Please delete the conclusion in the discussion section.

Conclusion

16. The conclusion is supported by findings and clearly answers the research question, while providing directions for future research. Thank you.  
